# Comparative Study of the Expression Profiles of miRNAs of Milk-Derived Exosomes of Yak and Jeryak

**DOI:** 10.3390/ani12223189

**Published:** 2022-11-17

**Authors:** Wenwen Ren, Yongfeng Zhang, Renqing Dingkao, Chun Huang, Xiaoming Ma, Xiaoyun Wu, Yongfu La, Min Chu, Pengjia Bao, Xian Guo, Jie Pei, Ping Yan, Chunnian Liang

**Affiliations:** 1Key Laboratory of Yak Breeding Engineering Gansu Province, Lanzhou Institute of Husbandry and Pharmaceutical Science, Chinese Academy of Agricultural Sciences, Lanzhou 730050, China; 2Key Laboratory of Animal Genetics and Breeding on Tibetan Plateau, Ministry of Agriculture and Rural Affairs, Lanzhou 730050, China; 3Animal Husbandry Station, Gannan Tibetan Autonomous Prefecture 747000, China

**Keywords:** yak, Jeryak, exosomes, miRNA

## Abstract

**Simple Summary:**

Exosomes are multivesicular bodies produced and secreted by living cells as membranous vesicles with a diameter of about 30–150 nm, density of 1.13–1.19 g/mL, and a typical “cup dish” shape. Exosomes carry proteins, miRNA, lncRNA, circRNA, and mRNA and their degradation fragments are involved in intracellular signal transduction important for regulation of cellular activities. Of these, miRNAs play an important role in the regulation of gene expression in important biological processes such as cell division, tissue development, and immune response. In this study, milk quality was determined and exosome miRNAs from yak and Jeryak milk sampled in June and December were characterized by sequencing. Differentially expressed miRNAs (DEMs) and pathways related to lactation performance were identified. In summary, this study provides data that can be used to develop alternative strategies for improving the lactation performance of yaks and Jeryaks.

**Abstract:**

The Jeryak is the hybrid offspring of yaks and Jersey cattle and exhibit improved milk and meat yields. Biomolecules carried within milk exosomes are important for cell growth, development, immune regulation, and various pathophysiological processes. Previous studies showed that miRNAs regulate mammary gland development, lactation, and milk quality. This study explored the relationship between milk exosomes miRNAs and lactation performance. A comparison of the milk content showed that yak milk was of a better quality compared to Jeryak milk (casein, fat, TS, SNF, lactose). Milk collected in December was superior to that collected in June for both yak and Jeryak, except for lactose concentrations. Exosomes were extracted by density gradient centrifugation and miRNA expression profiles in milk exosomes from three yaks and three Jeryaks collected in June and December were detected by small RNA sequencing. In all, 22, 120, 78, and 62 differentially expressed miRNAs (DEMs) were identified in Jun_ JY vs. Jun_ Y (P1: Jeryak in June vs. Yak in June), Jun_ JY vs. Dec_ JY (P2: Jeryak in June vs. Jeryak in December), Dec_ JY vs. Dec_ Y (P3: Jeryak in December vs. Yak in December), and Jun_ Y vs. Dec_ Y (P4: Yak in June vs. Yak in December) groups. These DEMs were enriched in functions and signaling pathways related to lactation performance. In conclusion, these findings are a reference tool to study the molecular basis of lactation performance.

## 1. Introduction

The yak (*Bos grunniens*) is a unique bovine species distributed in the alpine grassland of the Qinghai–Tibet Plateau and its adjacent areas at an altitude of more than 3000 m. Yaks are tolerant, have easily adapted to the low-oxygen environment of the plateau, and are a source of good-quality meat. The yak is also known as “boat of the plateau” or “multi-purpose livestock” [1]. By 2019, the number of yaks globally was about 17 million. China is the major breeder of yaks, with 16.21 million, or 95%, of the total number of yaks in the world [2]. Jersey cattle (*Bos taurus*) originated from Jersey in the British Channel Islands. Jersey cattle are relatively small in size; their most prominent feature is their high milkfat content [3]. The average annual milk output of the adult Jersey cow is about 4000 kg with a milkfat rate of 5%~6%, with some even reaching 7%~8% [4]. The Jeryak, a hybrid offspring of the yak (*Bos grunniens*) and Jersey cow (*Bos taurus*), demonstrates a higher level of heterosis, is larger in size, has a faster rate of growth and development, matures earlier, demonstrates higher feed utilization rate, has stronger adaptability to the ecological environment in alpine pastoral areas, has higher tolerance to roughage, and produces double the volume of milk and meat when compared with yaks [5].

Exosomes are cell-derived nanoscale extracellular vesicles (EV), which are vesicular structures surrounded by lipid bilayer membranes secreted by cells, and sized at 30–100 nm [6]. In 1987, these tiny vesicles, formed by vesicles of endosomes in the nucleus, were first named exosomes [7]. Most mammalian cells are capable of secreting exosomes and delivering their contents, which include proteins, nucleic acids, and lipids, to target cells, in turn affecting target-cell biological functions. Exosomes exist in various biological fluids such as blood, plasma, tears, semen, saliva, urine, cerebrospinal fluid, epididymal fluid, amniotic fluid, tumor ascites, bronchoalveolar lavage fluid, synovial fluid, milk, etc. Depending on their cellular origin, exosomes contain many cellular components, including mRNA, miRNA, rRNA, lncRNA, tRNA, DNA, lipids, metabolites, and proteins [8]. Exosomes are involved in regulating a variety of pathophysiological processes, such as immune inflammatory response, tumor growth, metabolism, and pregnancy [9]. Admyre et al. were the first to isolate and identify exosomes from human colostrum and standing milk [10]. Since then, exosomes have been reported in the colostrum and mature milk of humans, cows, buffaloes, goats, pigs, kangaroos, and rodents [11]. Milk exosomes can be absorbed and utilized by macrophages and vascular endothelial cells [12], are able to cross the blood–brain barrier, are bioavailable [13], and can reach peripheral tissues through the circulation of body fluids. Hence, milk exosomes have become the preferred vehicle for transporting biomolecules and regulators into the circulation of different organisms. Research has demonstrated that the various biomolecules carried by bovine milk exosomes play an important role in cell growth, development, immune regulation, and various pathophysiological processes [14,15,16,17].

MicroRNAs (miRNAs) are short noncoding (~22 nucleotides in length) regulatory RNAs that modulate gene expression at the post-transcriptional level, mostly by binding to perfectly/partially complementary sites at the 3′-UTR of target mRNAs [18,19]. MicroRNAs function to inhibit translation and expression of target genes (mRNAs) and are relatively conserved in biological evolution [20,21,22]. They are also involved in numerous cellular biological functions as natural protein translation regulators, and therefore miRNAs in exosomes have drawn much attention. In lactating and non-lactating bovine mammary gland tissues, miRNA transcriptome sequencing revealed that 56 of 921 miRNAs had distinct expression levels [23]. The miRNA expression profiles of mammary gland tissues during the dry period and peak lactation were evaluated, and certain miRNAs were found to potentially play roles in the growth or lactation processes of the mammary gland in dairy goats [24].

Yak milk is richer in proteins, essential amino acids, fat, lactose, and minerals compared with Holstein milk. The advantage of Jeryaks is obvious, as crossbreeding of the yak with Jersey cattle results in higher yields of milk compared to the yak: an economic advantage for herders. Studies have demonstrated that a variety of biomolecules carried by milk exosomes play important roles in cell growth, development, immune regulation, and various pathophysiological processes. However, no research has been done on the exosome miRNA expression profile between yaks and Jeryaks. Therefore, the screening of yak milk and Jeryak milk exosomes miRNAs can provide basic data to improve yak milk quality and yield and provide new information to guide subsequent breeding work.

## 2. Materials and Methods

### 2.1. Collection of Milk

Milk samples were collected in June and December from yaks (Jun_ Y and Dec_ Y) and Jeryaks (Jun_ JY and Dec_ JY) in Hezuo city, Gannan Tibetan Autonomous Prefecture, Gansu Province, China. Prior to collecting the milk, the udder was cleaned with warm water and then wiped thoroughly with a hot towel before milking. Samples of milk were collected from ten lactating yaks and Jeryaks (4 to 5 years old). The milk was transferred into 50 mL sterile centrifuge tubes, placed on dry ice, transported back to the laboratory, and immediately stored at −80 °C until used.

### 2.2. Determination of Milk Quality

Nutrient contents of the yak and Jeryak milk obtained in June and December were determined with the MilkoScanTM FT120 (Foss Analytical Instruments Co., Ltd., Danish). The contents tested included casein, protein, fat, total solids (TS), non-fat solid (SNF), lactose, freezing point, acidity, and citric acid content. SPSS24.0 *t*-test was utilized to detect the significance of any differences observed.

Data were divided into four groups for analysis: Jun_ JY vs. Jun_ Y (P1: Jeryak in June vs. Yak in June), Jun_ JY vs. Dec_ JY (P2: Jeryak in June vs. Jeryak in December), Dec_ JY vs. Dec_ Y (P3: Jeryak in December vs. Yak in December), and Jun_ Y vs. Dec_ Y (P4: Yak in June vs. Yak in December).

### 2.3. Extraction and Identification of Milk Exosomes from Yak and Jeryak Samples

#### 2.3.1. Extraction of Exosomes by Density Gradient Centrifugation

The milk samples were transferred into new centrifuge tubes and centrifuged at 4 °C, 2000× *g* for 30 min. The supernatant was carefully transferred to a fresh centrifuge tube and centrifuged again at 4 °C, 10,000× *g* for 45 min to remove the larger vesicles. The supernatant was transferred into a new centrifuge tube and centrifuge at 4 °C, 100,000× *g* for 120 min. The supernatant was decanted, and the precipitate was resuspended with 20 mL of pre-chilled 1×PBS and then centrifuged at 4 °C, 2000× *g* for 30 min. The supernatant obtained was centrifuged twice. After the final centrifugation step, the supernatant was decanted, and the pellet was resuspended with 1 mL of pre-chilled 1×PBS and stored temporarily at 4 °C until use. Iodixanol was prepared at different concentrations (40%, 20%, 10%, and 5%), and the different concentrations (3.6 mL each) were layered in the tube from high to low concentration. After the gradient had formed, 1 mL of the chilled suspension was added to the top layer. The gradient was then subjected to ultracentrifugation at 4 °C, 100,000× *g* for 120 min. After centrifugation, 12 layers were noted and the middle 6–9 layers of liquid were ultracentrifuged at 4 °C, 100,000× *g* for 120 min. The supernatant was removed, and the pellet was resuspended with 300 μL pre-chilled 1 × PBS. Then, 20 μL was used for electron microscopy, 10 μL for pellet size determination, and the remaining exosomes were stored at −80 °C.

#### 2.3.2. Identification of Exosome

The exosome morphology was observed using transmission electron microscopy (HT-7700, Hitachi Ltd., Tokyo, Japan). The exosome particle size was analyzed with a particle size analyzer (N30E, Xiamen Fuliu Biotechnology Co., Ltd., Xiamen, China).

### 2.4. Extraction of Exosome RNA

Extraction of exosome RNA from the milk obtained from yaks and Jeryaks in June and December was performed with the Norgen 58,000 kit (Norgen Bio Inc., Ontario, Canada) based on the manufacturer’s instructions.

### 2.5. Construction and Sequencing of Small RNA Library

The experimental procedure was performed according to the standard steps provided by Illumina, including library preparation and sequencing experiments. Small RNA sequencing libraries were prepared using the TruSeq Small RNA Sample PrepKits (Illumina, San Diego, CA, USA) kit. After library preparation, the constructed libraries were sequenced using the Illumina Hiseq2000/2500 platform with the single-end 1 × 50 bp read length strategy.

### 2.6. Preprocessing of Sequencing Data

Raw reads were subjected to an in-house program, ACGT101-miR (LC Sciences, Houston, Texas, USA), to remove adapter dimers, junk, low complexity reads, common RNA families (rRNA, tRNA, snRNA, snoRNA), and repeats. Subsequently, unique sequences of 18~26 nucleotide were mapped to specific species precursors in the miRBase 22.0 using a BLAST search to identify known miRNAs and novel 3p- and 5p-derived miRNAs. Length variation at both 3′ and 5′ ends and one mismatch within the sequence were allowed in the alignment. The unique sequences mapping to specific species mature miRNAs in hairpin arms were identified as known miRNAs. The unique sequences mapping to the other arm of known specific species precursor hairpins opposite to the annotated mature miRNA-containing arm were novel 5p- or 3p-derived miRNA candidates. The remaining sequences were mapped to other selected species precursors (with the exclusion of specific species) in miRBase 22.0 by a BLAST search, and the mapped pre-miRNAs were further BLASTed against the specific species genomes to determine their genomic locations. The above two are defined as known miRNAs. The unmapped sequences were BLASTed against the specific genomes and the hairpin RNA structures containing sequences were predicated from the flanking 80 nt sequences using the RNAfold software (http://rna.tbi.univie.ac.at/cgi-bin/RNAfold.cgi. accessed on 3 July 2022). The criteria for secondary structure prediction were (1) number of nucleotides in one bulge of the stem is ≤12; (2) number of base pairs in the stem region of the predicted hairpin ≥16 nt; (3) cutoff of free energy was set at ≤−15 kCal/mol; (4) length of hairpin (including up and down stems and the terminal loop) is ≥50 nt; (5) length of hairpin loop is ≤20 nt; (6) number of nucleotides in one bulge in one bulge within the mature region must be ≤8 nt; (7) number of biased errors in one bulge within the mature region is ≤4; (8) the number of biased bulges in mature region is ≤2; (9) number of errors in mature region is ≤7; (10) number of base pairs in the mature region of the predicted hairpin is set at ≥12; (11) percent of mature miRNAs within the stem structure is ≥80%.

### 2.7. Differential miRNA Expression Analysis

The expression of miRNA was normalized with the ACGT101-miR software, and the norm value (equivalent to the FPKM value used in transcriptome analysis) was obtained. Then, the miRNA sequencing expression profiles were normalized as follows: First, a common set of sequences was identified among all samples to construct a reference dataset with each data point in the reference set representing the copy number median value of a corresponding common sequence of all samples. Then, 2-based logarithm transformation was performed on copy numbers ($\log_{2}\left(copy\#\right)$) of all samples and reference datasets. The $\log_{2}\left(copy\#\right)$ difference ($\Delta \log_{2}\left(copy\#\right)$) between individual sample and the reference data set was calculated. A subset of sequences was created by selecting $\;\left|\Delta \right.\log_{2}(copy\#\left.)\right|<2$, which indicated less than ($2^{2}=$) 4-fold change from the reference set. Linear regressions were performed on the subset sequences between individual samples and the reference set to derive the linear equation $y=a_{i}x+b_{i}$, where $a_{i}$ and $b_{i}$ are the slop and interception, respectively, of the derived line, x is $\log_{2}\left(copy\#\right)\;$ of the reference set, and y is the expected $\log_{2}\left(copy\#\right)$ of sample i on a corresponding sequence. The mid value $\;\;x_{mid}=\frac{\max\left(x\right)-\min\left(x\right)}{2}$ of the reference set was calculated. Then, the corresponding excepted $\log_{2}\left(copy\#\right)$ of sample i $y_{i,mid}=a_{i}x_{mid}+b_{i}$. Let $y_{r,mid}=x_{mid}$. Let $\Delta y_{i}=y_{r,mid}-y_{i,mid}$, which is the logarithmic correction factor of sample i. Then, the arithmetic correction factor $f_{i}=2^{\Delta y_{i}}$ of sample i was derived. Copy numbers of individual samples were corrected by multiplying corresponding arithmetic correction factor $f_{i}\;$ with the original copy numbers.

The input data for DEMs analysis were normalized data (norm values), and the *p*-value calculation model based on the normal distribution was used for *p*-value calculations, and a *t*-test was used to analyze the differences between the two groups of samples. The DEMs were identified based on $\;\left|\log_{2}(\mathrm{fold}\;\mathrm{change}\left.)\right|\geq 1\right.$ and *p* < 0.05.

### 2.8. Prediction of Target Genes and GO, KEGG Enrichment Analysis

Targetscan (V5.0) and Miranda (v3.3a) software were used to predict target genes of the miRNAs that were differentially regulated. The target genes predicted by the two software were screened according to the scoring criteria of each software. ThTargetscan algorithm removed target genes with a context score percentage of less than 50, and the Miranda algorithm removed target genes with maximum free energy greater than −10 (i.e., the threshold was Targetscan score ≥ 50, Miranda_ energy < −10). The intersection of software results was taken as the final set of genes targeted by the differentially expressed miRNAs. The results were analyzed for functional enrichment using te Gene Ontology (GO) (http://www.geneontology.org/. accessed on 5 July 2022) and KEGG Pathway (http://www.genome.jp/kegg/. accessed on 5 July 2022) databases.

## 3. Results

### 3.1. Determination of the Yak and Jeryak Milk Quality

In the P1 group (Table 1), yak milk casein, fat, TS, SNF, lactose, and freezing point were significantly higher than Jeryak milk (*p* < 0.01). For the P2 group (Table 2), casein, protein, TS, freezing point, and acidity of the Jeryak milk in June were significantly lower than that in the samples collected in December (*p* < 0.01), while fat levels were also significantly lower when compared to milk harvested in December (*p* < 0.05). On the other hand, lactose and citric acid were significantly higher in the June milk samples compared to the December sample (*p* < 0.05). The P3 group (Table 1) of Yak milk had freezing points and citric acid levels significantly higher than the Jeryak milk (*p* < 0.01), with SNF also significantly higher than the Jeryak milk (*p* < 0.05). In the P4 group (Table 2), the protein content and acidity of the yak milk sampled in June were significantly lower than in December (*p* < 0.01), whereas lactose levels were significantly higher than milk sampled in December (*p* < 0.01).

### 3.2. Identification of Milk Exosomes from Yak and Jeryak Milk Collected in June and December

The exosomes of yak and Jeryak are spherical with a depression in the middle, which conforms to the typical appearance of exosomes (Figure 1). The milk exosomes particle sizes for yak and Jeryak milk obtained in June and December ranged between 30–150 nm (Figure 2). The mean particle sizes and concentration of exosomes for yak and Jeryak milk sampled in June and December are shown in Table 3.

### 3.3. Characteristic Analysis of Yak and Jeryak Milk Exosome miRNAs Obtained in June and December

Statistical analysis was performed on the output of the original sequencing data to obtain the unique sequences within the sequencing data and the corresponding copy number of each unique sequence. Initially, quality control was performed on the original sequencing data to removes the 3′ junction sequence and sequences of less than 18 nt in length. Sequences that contained 80% A or C or G or T, 3N (not necessarily continuous), only A and C with no G and T, or only G and T with no A and C, or continuous nucleotide dimers and trimers were removed. At the same time, we compared and filtered the sequences against the mRNA (which may not be available in some species), RFam (including rRNA, tRNA, snRNA, snoRNA, etc.), and Repbase databases. We referred to this filtered data as valid data. Further miRNA identification and prediction analysis were performed on the valid data (Appendix A). The length of identified miRNAs was determined by assuming that the horizontal coordinate is the miRNA length and the vertical coordinate is the percentage of de-duplicated miRNAs (reason for de-duplication: the precursor of a miRNA is simultaneously compared to two positions of the genome). The miRNA length distribution was mainly between 18–24 nt, with 19 nt accounting for the largest proportion (Appendix A). The sequence length is typical of Dicer enzyme cleavage and reflects the accuracy of the miRNA sequencing.

### 3.4. Analysis of Differential miRNAs

A total of 22 DEMs were identified in group P1, of which 14 were upregulated and 8 were downregulated. A total of 120 DEMs were identified in group P2, of which 51 were upregulated and 69 were downregulated. In group P3, 78 DEMs were identified, of which 55 were upregulated and 23 were downregulated. In group P4, 62 DEMs were identified (21 upregulated and 41 downregulated DEMs). (Appendix A). The volcano plots of all four groups are shown in Figure 3. The heat map of DEMs of all four groups are shown in Appendix A.

### 3.5. Enrichment Analysis of Target Genes

To further investigate the biological functions regulated by the DEMs, the target genes of the significantly DEMs were predicted separately using TargetScan and Miranda. The intersection of the two software was taken as the final list of target genes. The results of the top 20 enriched GO terms and KEGG pathways showed that the four groups were mainly enriched for extracellular exosome, zinc ion binding, cytoplasm, etc. KEGG functional analysis of the identified target genes showed that they were mainly enriched in Axon guidance, MAPK signaling pathway, and Ras signaling pathway, etc. (Figure 4).

## 4. Discussion

Yak is a unique livestock resource in the Qinghai–Tibet Plateau and its adjacent areas in China. The yak is the only breed of cattle that is able to take advantage of the alpine grasslands of the Qinghai–Tibet Plateau at an altitude of over 3000 m for animal production. Yak milk is characterized by its high content of dry matter and fat: both colostrum and regular milk have a higher content of dry matter, milk fat, protein, lactose, and other nutrients than cow’s milk, and yak milk has large fat globules (10.28 μm) with high milk fat content. Yak milk is the ideal source of raw milk to be processed into cream and its products [25]. Compared to butter from cow, sheep and goat, butter from yak has higher contents of conjugated linoleic acid, monounsaturated fatty acids, oleic acid, and n-3 long-chain polyunsaturated fatty acids and the lowest content of saturated fatty acids [26]. Under the same feeding conditions, cattle x yak hybrid milk yield is significantly higher than that of yak, but the milk fat and dry matter content of cattle x yak hybrid milk are lower than that of yak milk [27]. Our characterization confirms these previous findings. The lactose content of yak and Jeryak milk obtained in December was lower than that obtained in June. During winter in the alpine areas, yaks need more energy to keep warm while grass withers and turns yellow, cellulose content increases, and nutrient content decreases [28].

Milk contains a large number of exosomes that, because of their unique bilayer membrane structure, can be used as drug delivery carriers, alleviate inflammation, and have important applications and research value as biomarkers of the health status of different organisms [29]. Exosome extraction methods include ultra-high-speed centrifugation, density gradient centrifugation, and the use of specific kits. However, due to the high abundance of casein in yak milk, it is difficult to isolate and purify yak milk exosomes [30]. In this study, density gradient centrifugation was used to extract the milk exosomes of yak and Jeryak, and the appearance as well as particle size was consistent with the typical characteristics of exosomes. Gao [31] found that the rennet-assisted optimization method (ultra-high-speed centrifugation + rennet precipitation) can effectively extract exosomes from bovine or Yak milk. The principle is based on the effective binding of rennet to casein in cow’s milk, which facilitates the removal of casein from cow’s milk.

Further, miRNAs are involved in the regulation of lactation signaling pathways, physiological processes, health [32], and the properties of milk [33,34]. For example, miR-100 and miR-146b target mammary gland metabolism via the mTOR signaling pathway, and their differential expression correlated with milk yield. Similarly, miR-100, miR-30e-5p, miR-25, and miR-16a target lipid metabolism and promote milk fat synthesis [35]. In this study, we compared the expression profiles of yak and Jeryak exosome DEMs. We found that most of the DEMs were enriched for the same functions and signal pathways, such as extracellular exosomes, zinc ion binding, protein binding, axon guidance, MAPK signaling pathway, Ras signaling pathway, and other signaling pathways related to lactation and milk quality. Studies have shown that the PI3K–AKT signaling pathway interacts with the JAK–STAT signaling pathway and MAPK signaling pathway, both of which play an important role in regulating the synthesis and secretion of milk components and other biological processes [36]. Moreover, miRNA plays an important regulatory role in mammary gland milk fat synthesis. 

In this study, miRNAs associated with mammary gland milk fat were identified among the four groups of DEMs. Overexpression of miR-130a significantly decreased the levels of triacylglycerol (TAG) and inhibited the formation of lipid droplets, while inhibition of miR-130a led to more lipid droplet formation and TAG accumulation in bovine mammary epithelial cells (BMEC). This suggests that miR-130a may be used to improve the beneficial constituents of cow milk [37]. In addition, bta-miR-199c may be an important regulator of adipogenesis and metabolism in the bovine mammary gland [38]. In another study, bta-miR-146b was upregulated in Holstein cows at 30 days of gestation under the hypotheses that it is related to mammary gland development [39] and is a potential regulator of fatty acid synthesis [40]. Finally, bta-miR-26b plays an important role in the regulation of cell biology processes, whereby it participates in the early development of embryos, regulates cell proliferation, and affects pituitary hormone secretion and other physiological activities, including those related to reproductive functions [41]. 

The expression of miR-26 and its target gene family (carboxy-terminal domain RNA polymerase II polypeptide A small phosphatase, CTDSP) was significantly higher in dairy goats during the lactation period compared to the dry period. Expression levels peaked at the full lactation period, and CTDSP was moderately correlated with the content of total fat solids in the milk [42]. The role of target interleukin-1 (IL-1) in regulating the mammary gland function of dairy cows may be affected by bta-miR-181d and bta-let-7c [43]. The miRNAs mentioned above have been proven to play an important role in the lactation process. A previous study demonstrated that different miRNAs expression profiles were observed in Sahiwal (Bos indicus) and Frieswal (*Bos indicus* × *Bos Taurus*) under summer stress conditions. When compared to Sahiwal, Frieswal bta-miR-2478 was significantly up-regulated in the summer (*p* < 0.01) [44]. On the other hand, over the winter period, expression of bta-let-7b and bta-miR-16a was significantly lower than in the summer (*p* < 0.01) [45]. 

Our study also compared the expression of exosome miRNAs in yaks and Jeryaks over different seasons and noted that our transcriptome analysis concurred with the previous reports. In addition, we also analyzed the expression of exosome miRNAs in yaks and Jeryaks in the same season. We noted that mammary gland development potential for beef cattle differed from dairy cattle and mammary gland development intensity and that productivity of beef cattle breeds was much less [46]. These differences may result from different levels of genetic and central endocrine regulation, but may also depend on intramammary factors, including the number of mammary gland stem cells and gene expression involved in the development of the mammary gland stem cell niche [47].

## 5. Conclusions

This study is the first to extract and identify milk exosomes from yaks and Jeryaks over different seasons. The animals’ lactation performance was established, while the exosomes miRNA expression profiles were compared between animal species and different seasons. From the analysis, miRNAs and pathways related to lactation performance were identified. We propose to extend this study by selecting significant DEMs for functional verification to identify the key miRNAs affecting lactation performance. Collectively, the data from these studies will provide a theoretical basis for improving the lactation performance of yaks and Jeryaks.

## Figures and Tables

**Figure 1 animals-12-03189-f001:**
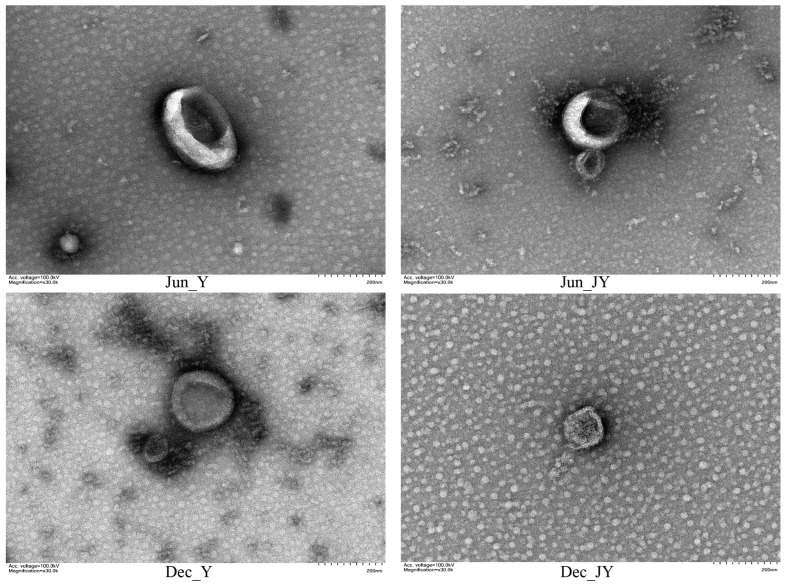
Milk exosomes from yak and Jeryak sampled in June and December as determined by transmission electron microscopy.

**Figure 2 animals-12-03189-f002:**
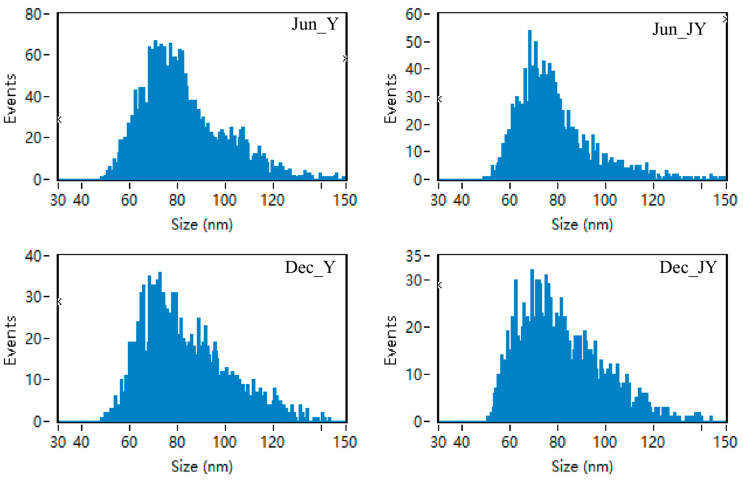
Range of milk exosomes particle sizes for yak and Jeryak milk collected in June and December.

**Figure 3 animals-12-03189-f003:**
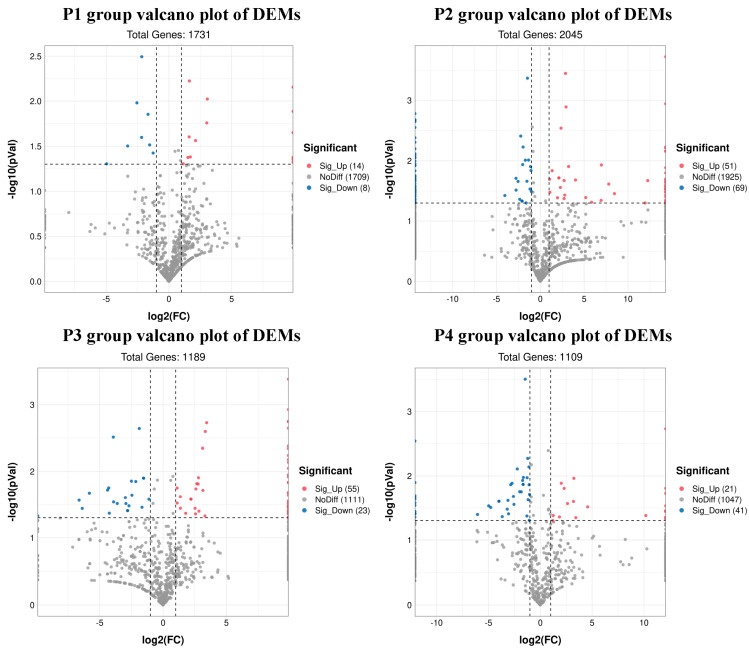
Volcano plots showing differentially expressed miRNAs (red and blue dots).

**Figure 4 animals-12-03189-f004:**
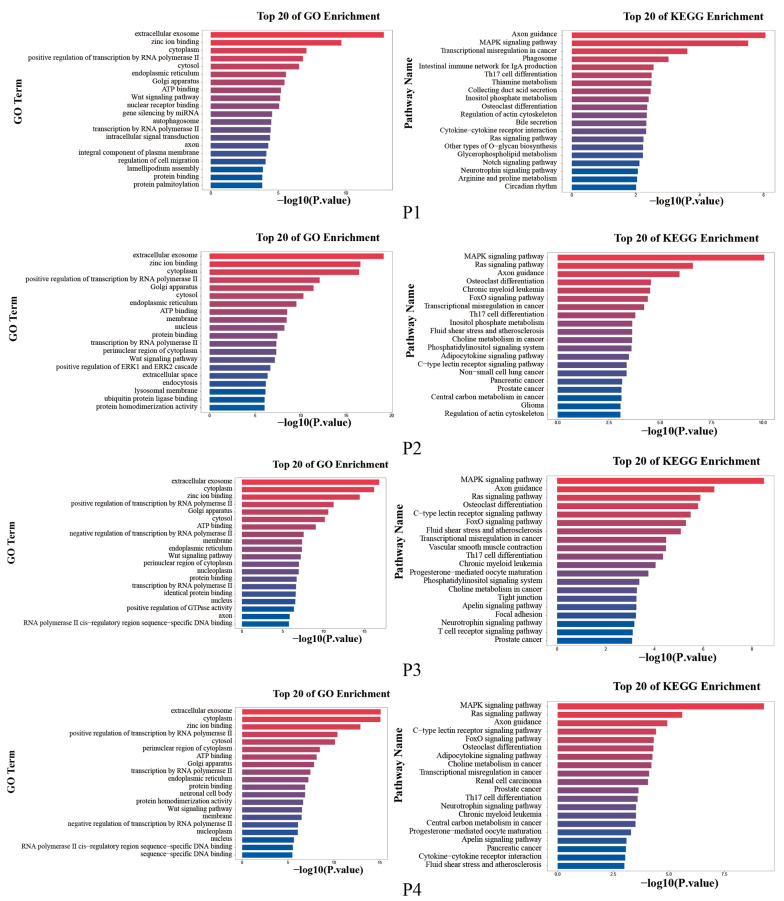
The top 20 enriched GO terms and KEGG pathways.

**Table 1 animals-12-03189-t001:** Comparison between the milk quality of yak and Jeryak milk sampled in June (P1: Jeryak in June vs. Yak in June) and December (P3: Jeryak in December vs. Yak in December).

	June (P1)				December (P3)			
Milk Quality	Yak	Jeryak	Significance	*p*-Value	Yak	Jeryak	Significance	*p*-Value
Casein	3.739 ± 0.211	3.189 ± 0.209	**	<0.01	4.049 ± 0.692	3.754 ± 0.348		0.284
Protein	4.056 ± 0.206	3.831 ± 0.322		0.071	5.011 ± 0.927	4.746 ± 0.691		0.476
Fat	8.137 ± 2.095	4.670 ± 0.930	**	<0.001	7.399 ± 2.726	6.164 ± 1.471		0.241
TS	18.550 ± 1.950	14.367 ± 0.877	**	<0.001	18.195 ± 3.364	16.296 ± 1.782		0.146
SNF	10.402 ± 0.280	9.834 ± 0.338	**	<0.001	10.779 ± 0.630	10.094 ± 0.521	*	0.012
Lactose	5.275 ± 0.092	4.829 ± 0.180	**	<0.001	4.600 ± 0.340	4.333 ± 0.234		0.054
Freezing point	0.700 ± 0.000	0.600 ± 0.000	**	<0.001	0.737 ± 0.060	0.663 ± 0.052	**	0.005
Acidity	8.886 ± 0.788	8.953 ± 1.020		0.866	13.110 ± 2.523	11.625 ± 1.877		0.148
Citric acid	0.209 ± 0.221	0.208 ± 0.256		0.921	0.214 ± 0.437	0.166 ± 0.303	**	0.009

Note: ** *p* < 0.01, * *p* < 0.05.

**Table 2 animals-12-03189-t002:** Comparison between the quality of milk sampled in June and December from Jeryak milk (P2: Jeryak in June vs. Jeryak in December) and Yak milk (P4: Yak in June vs. Yak in December).

	Jeryak (P2)				Yak (P4)			
Milk Quality	June	December	Significance	*p*-Value	June	December	Significance	*p*-Value
Casein	3.189 ± 0.209	3.754 ± 0.348	**	0.002	3.739 ± 0.211	4.0489 ± 0.692		0.181
Protein	3.831 ± 0.322	4.746 ± 0.691	**	0.001	4.056 ± 0.206	5.0105 ± 0.927	**	0.004
Fat	4.670 ± 0.930	6.164 ± 1.471	*	0.012	8.137 ± 2.095	7.3989 ± 2.726		0.462
TS	14.367 ± 0.877	16.296 ± 1.782	**	0.005	18.550 ± 1.950	18.1953 ± 3.364		0.762
SNF	9.834 ± 0.338	10.094 ± 0.521		0.191	10.402 ± 0.280	10.7789 ± 0.630		0.085
Lactose	4.829 ± 0.180	4.333 ± 0.234	**	<0.001	5.275 ± 0.092	4.5995 ± 0.340	**	<0.001
Freezing point	0.600 ± 0.000	0.663 ± 0.052	**	<0.001	0.700 ± 0.000	0.7368 ± 0.060		0.064
Acidity	8.953 ± 1.012	11.625 ± 1.877	**	0.001	8.886 ± 0.788	13.1095 ± 2.523	**	<0.001
Citric acid	0.208 ± 0.256	0.166 ± 0.303	**	0.004	0.209 ± 0.221	0.2140 ± 0.437		0.723

Note: ** *p* < 0.01, * *p* < 0.05.

**Table 3 animals-12-03189-t003:** Milk exosomes particle size and concentration of yak and Jeryak sampled in June and December.

Sample Name	Mean Particle Size (nm)	Concentration (Particles/mL)
Jun_ Y	81.95	3.85 × 10^10^
Jun_ JY	77.80	2.05 × 10^10^
Dec_ Y	82.61	2.59 × 10^10^
Dec_ JY	80.82	6.89 × 10^9^

## Data Availability

The data were submitted to the database of the Gene Expression Omnibus (GEO). The accession number is GSE212938.

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
