# Peer review of "Comparative Study of the Expression Profiles of miRNAs of Milk-Derived Exosomes of Yak and Jeryak"

_animals, 2022, doi:10.3390/ani12223189_

Round 1

Reviewer 1 Report

In this manuscript, Ren et al. attempted to compare the miRNA profiles of Yak and Jersey-Yak cattle breeds using miRNA-seq data. The results from this study can be used to improve the understanding of the molecular basis of lactation performance of these breeds. Some comments:

1-    Lines 160-166: The name of the software(s)/quality criteria used for miRNA-read cleaning and mapping to the genome needs to be included in the method section.

2-    Lines 167-176: were miRNA target sites predicted on genes 3’UTR or gene body? Also, were non-coding genes included in the analysis or not?

Author Response

Dear Reviewer,

Thank you for your useful comments and suggestions on the language and the structure of our manuscript. We have modified the manuscript accordingly, and the detailed corrections are listed below point by point:

  1. Lines 160-166: The name of the software(s)/quality criteria used for miRNA-read cleaning and mapping to the genome needs to be included in the method section.

Response: Raw reads were subjected to an in-house program, ACGT101-miR (LC Sciences, Houston, Texas, USA) to remove adapter dimers, junk, low complexity, common RNA families (rRNA, tRNA, snRNA, snoRNA) and repeats. Subsequently, unique sequences with length in 18~26 nucleotide were mapped to specific species precursors in miRBase 22.0 by BLAST search to identify known miRNAs and novel 3p- and 5p- derived miRNAs. Length variation at both 3’ and 5’ ends and one mismatch inside of the sequence were allowed in the alignment. The unique sequences mapping to specific species mature miRNAs in hairpin arms were identified as known miRNAs. The unique sequences mapping to the other arm of known specific species precursor hairpin opposite to the annotated mature miRNA-containing arm were considered to be novel 5p- or 3pderived miRNA candidates. The remaining sequences were mapped to other selected species precursors (with the exclusion of specific species) in miRBase 22.0 by BLAST search, and the mapped pre-miRNAs were further BLASTed against the specific species genomes to determine their genomic locations. The above two we defined as known miRNAs. The unmapped sequences were BLASTed against the specific genomes, and the hairpin RNA structures containing sequences were predicated from the flank 80 nt sequences using RNAfold software (http://rna.tbi.univie.ac.at/cgi-bin/RNAfold.cgi). The criteria for secondary structure prediction were: (1) number of nucleotides in one bulge in stem (≤12). (2) number of base pairs in the stem region of the predicted hairpin (≥16). (3) cutoff of free energy (kCal/mol ≤-15). (4) length of hairpin (up and down stems + terminal loop ≥50). (5) length of hairpin loop (≤20). (6) number of nucleotides in one bulge in mature region (≤8). (7) number of biased errors in one bulge in mature region (≤4). (8) number of biased bulges in mature region (≤2). (9) number of errors in mature region (≤7). (10) number of base pairs in the mature region of the predicted hairpin (≥12). (11) percent of mature in stem (≥80).

  1. Lines 167-176: were miRNA target sites predicted on genes 3’UTR or gene body? Also, were non-coding genes included in the analysis or not?

Response: miRNA target sites predicted on genes 3’UTR, and non-coding genes were not included in the analysis

Reviewer 2 Report

The manuscript provided some novel information about the roles of miRNAs in yak and Jersey-jak. Overall, the methods are reasonable, and the approach is fine. The authors provided some justifications for methods and approaches. The introduction is enough but might need to focus on the roles of miRNAs in lactation and milk regulations (which are known and unknown) then introduce the miRNA’s biogenesis. The authors should redo the statistical test for each milk composition. Details about the processing of sequence and the DE analyses need to be provided. Moreover, the authors focus more on the DE miRNAs and these functions for both results and discussion, rather than results of enriched pathways from their potential target genes.

  Line 21: Define the abbreviation for DEM,

Line 26-27: “December was significantly better than that of June “ for which group, yak or jersey-yak?

Line 28: Should give the number of samples

Change high-throughput sequencing technology to RNA sequencing or small RNA sequencing

Should have one sentence about the results of DEM before talking about the pathways and genes? The DEM is a more important result.

Line 77-96: The authors might not need to provide details about the biogenesis of miRNAs but could replace this text with more functions of miRNAs in lactation/milk genesis and regulation.

The authors should justify why choosing December and June for experiments. Are diets different in these periods? What are the basic diets for yak and jersey-jak?

Immortally, the miRNA expression varies in the lactation cycle. Why did the authors choose 120 days (mid-lactation) for collecting the mink samples for exosome isolation and later miRNA sequencing?

At which time of the day and which quarter of milk was collected?

Line 124-125: The authors should perform multiple comparisons to see if the milk composition is affected by  “breed” and “time”, all four tables should be joined into one table and a posthoc test should be done.

What is the quality of RNAs after isolation?

Line 152: Add the company information

More information about the programs and bioinformatic pipelines for processing miRNAs should be provided.

Line 169: Where is the information about DE analyses?

Why did the authors choose target scan version 5, it is very outdated.

The sequence should be deposited

For tables, the authors should use a maximum of three digitals

Figure 3 and table 6 could move to supplementary files

Figure 4: Should change the titles for each sup-figure in the files.

The heatmap is not clear and might not be necessary. Most information about the DEMs should be presented well in the tables in the main manuscript. Other pathways and GO analyses are supporting information.

Are these pathways significantly enriched? The authors could add only the significantly enriched GO and pathways, and also need to enhance the quality of the figure.

Line 296: What did the authors mean by “differential miRNAs”, please use the abbreviations when possible.

Line 315: Which species is for miR-26?

The conclusion does not guide the readers on what is new and what should be done next. 

Author Response

Dear Reviewer,

Thank you for your useful comments and suggestions on the language and the structure of our manuscript. We have modified the manuscript accordingly, and the detailed corrections are listed below point by point:

  1. Line 21: Define the abbreviation for DEM,

Response: Have defined the abbreviation for Differential express miRNAs (DEMs).

  1. Line 26-27: “December was significantly better than that of June “for which group, yak or jersey-yak?

Response: Have changed the sentence “the December was significantly better than that of June of yak and Jersey Cattle-yak except lactose.” In addition to lactose, whether yak or Jersey Cattle-yak, the quality of milk in December is better than that in June, because the milk yield in December is relatively less than that in June, therefore, the content is relatively high. The lactose of yak and Jersey cattle-yak in December were lower than that in June, in the winter of alpine areas, yaks need more energy, while grass withers and turns yellow, cellulose content increases and nutrients decrease.

  1. Line 28: Should give the number of samples

Response: Added the number of samples: Exosomes were extracted by density gradient centrifugation, and miRNA expression profiles of milk exosomes from three Yak and Jersey Cattle-yak in June and December were detected by small RNA sequencing technology.

  1. Change high-throughput sequencing technology to RNA sequencing or small RNA sequencing

Response: “high-throughput sequencing technology” was replaced by “small RNA sequencing technology”

  1. Should have one sentence about the results of DEM before talking about the pathways and genes? The DEM is a more important result.

Response: Line29-30: Added the results of DEM: 22, 120, 78, and 62 DEMs were identified in P1-P4 groups.

  1. Line 77-96: The authors might not need to provide details about the biogenesis of miRNAs but could replace this text with more functions of miRNAs in lactation/milk genesis and regulation.

Response: It has been modified and add relevant references. The production of animal milk is a complex biological process that is precisely regulated by many gene networks. Although studies have reported that many genes are involved in the biosynthesis process of breast milk, and miRNA sequencing results indicate that some mi RNAs may be involved in the formation process of milk, the relevant mechanism is not clear.

  1. The authors should justify why choosing December and June for experiments. Are diets different in these periods? What are the basic diets for yak and jersey-yak?

Response: Generally speaking, June is the peak period of milk production due to the vigorous growth of herbage, while December is the time of depletion of herbage and the decrease of milk yield. These two times are chosen because of the large time interval and significant changes in milk yield. Because the milk is collected from herdsmen's yaks and Jersey cattle-yak, their diet is basically the same. They are naturally grazed in June and grazing plus supplement feeding in December.

  1. Immortally, the miRNA expression varies in the lactation cycle. Why did the authors choose 120 days (mid-lactation) for collecting the milk samples for exosome isolation and later miRNA sequencing?

Response: After calving, the milk yield increases with time, reaching the peak of milk yield around 120 days, and then the milk yield decreases with time. The sentence has been modified: 10 samples of milk from each 4–5-year-old, 2-3 lactating yak and Jersey cattle-yak (120±3d postpartum) were collected in June, and collected again in December.

  1. At which time of the day and which quarter of milk was collected?

Response: Generally, herdsmen collect milk twice a day in the morning and evening, and we choose to collect milk at 6:00 a.m.

  1. Line 124-125: The authors should perform multiple comparisons to see if the milk composition is affected by “breed” and “time”, all four tables should be joined into one table and a posthoc test should be done.

Response: Thanks for your suggestion, however, T-test was used in this study, and multiple comparisons can only be made when there are more than three groups. In our study, only two groups be compared, and the previous materials and methods have been modified. In addition, these four tables have been joined into two tables, while would be too large to joined into one table.

  1. What is the quality of RNAs after isolation?

Response: Requirements for animal tissue samples: RIN>6.3, However, serum, plasma, and exosomes do not require RIN values, due to the small fragment of miRNA, it is less affected by degradation.

  1. Line 152: Add the company information

Response: Add the company information: Norgen 58000 kit (Canada, Norgen)

  1. More information about the programs and bioinformatic pipelines for processing miRNAs should be provided.

Response: Raw reads were subjected to an in-house program, ACGT101-miR (LC Sciences, Houston, Texas, USA) to remove adapter dimers, junk, low complexity, common RNA families (rRNA, tRNA, snRNA, snoRNA) and repeats. Subsequently, unique sequences with length in 18~26 nucleotide were mapped to specific species precursors in miRBase 22.0 by BLAST search to identify known miRNAs and novel 3p- and 5p- derived miRNAs. Length variation at both 3’ and 5’ ends and one mismatch inside of the sequence were allowed in the alignment. The unique sequences mapping to specific species ma-ture miRNAs in hairpin arms were identified as known miRNAs. The unique sequenc-es mapping to the other arm of known specific species precursor hairpin opposite to the annotated mature miRNA-containing arm were considered to be novel 5p- or 3pderived miRNA candidates. The remaining sequences were mapped to other select-ed species precursors (with the exclusion of specific species) in miRBase 22.0 by BLAST search, and the mapped pre-miRNAs were further BLASTed against the specific spe-cies genomes to determine their genomic locations. The above two we defined as known miRNAs. The unmapped sequences were BLASTed against the specific ge-nomes, and the hairpin RNA structures containing sequences were predicated from the flank 80 nt sequences using RNAfold software (http://rna.tbi.univie.ac.at/cgi-bin/RNAfold.cgi). The criteria for secondary structure prediction were: (1) number of nucleotides in one bulge in stem (≤12). (2) number of base pairs in the stem region of the predicted hairpin (≥16). (3) cutoff of free energy (kCal/mol ≤-15). (4) length of hair-pin (up and down stems + terminal loop ≥50). (5) length of hairpin loop (≤20). (6) num-ber of nucleotides in one bulge in mature region (≤8). (7) number of biased errors in one bulge in mature region (≤4). (8) number of biased bulges in mature region (≤2). (9) number of errors in mature region (≤7). (10) number of base pairs in the mature region of the predicted hairpin (≥12). (11) percent of mature in stem (≥80).

  1. Line 169: Where is the information about DE analyses?

Response: Line184-186: added the information about DE analyses

  1. Why did the authors choose target scan version 5, it is very outdated.

Response: For the targetscan software, we use the local script of the targetscan software. We use 3'UTR and miRNA to make predictions based on the prediction principle of targetscan, instead of using web pages. Currently, targetscan has been updated to version 8.0, but the local version of 8.0 is very memory consuming and runs very slowly. We have also done tests. The difference between versions 5.0 and 8.0 is very small, So the current version is still 5.0

  1. The sequence should be deposited

Response: The data were submitted to the data base of the Gene Expression Omnibus (GEO). The appropriate number for accession is GSE212938.

  1. For tables, the authors should use a maximum of three digitals

Response: The table has been changed to keep three decimal places.

  1. Figure 3 and table 6 could move to supplementary files

Response: Figure 3 and table 6 has moved to supplementary files.

  1. Figure 4: Should change the titles for each sup-figure in the files.

Response: Has changed the titles for each sup-figure in the files.

  1. The heatmap is not clear and might not be necessary. Most information about the DEMs should be presented well in the tables in the main manuscript. Other pathways and GO analyses are supporting information.

Response: The heatmap has moved to supplementary files. And the information about the DEMs was in supplementary files.

  1. Are these pathways significantly enriched? The authors could add only the significantly enriched GO and pathways, and also need to enhance the quality of the figure.

Response: These pathways are significantly enriched, line 269 has added the sentence “The results of top 20 GO and KEGG enrichment showed…”. And the quality of the figure has been enhanced.

  1. Line 296: What did the authors mean by “differential miRNAs”, please use the abbreviations when possible.

Response: “differential miRNAs” was replaced by “DEMs”

  1. Line 315: Which species is for miR-26?

Response: A PhD dissertation on miR-26 in dairy goats was referred, added the species “dairy goats” in line 316.

  1. The conclusion does not guide the readers on what is new and what should be done next.

Response: This study was the first extracted and identified the milk exosomes of yaks and Jersey Cattle-yak in June and December, determined their lactation performance, and compared their exosomes miRNA expression profiles by small RNA sequencing technology, and identified miRNAs and pathways related to lactation performance. Next, we will select significantly DEMs for functional verification to identify the key miRNAs affecting lactation performance. This study can provide a theoretical basis for improving the lactation performance of yaks and Jersey Cattle-yaks.

Round 2

Reviewer 2 Report

Thank you for providing responses. All my comments have been addressed. 

Some minor points:

Change Differential to differential in line 21.

Line 25-26: Should clarify which is milk quality, and add the indicators for it.

Line 30: P1-P4, should be defined before using. Otherwise, the authors should write the full group name.

Line 186-187: it is unclear about the DE analyses, which software was used, and which models were used.

Line 201-204: It should be in the method section, not in the results
